# Three-Dimensional Characterization of a Coastal Mode-Water Eddy from Multiplatform Observations and a Data Reconstruction Method

Ivan Manso-Narvarte [1,*], Anna Rubio [1], Gabriel Jordà [2], Jeffrey Carpenter [3], Lucas Merckelbach [3] and Ainhoa Caballero [1]

1  AZTI, Marine Research, Basque Research and Technology Alliance (BRTA), Herrera Kaia, Portualdea z/g, 20110 Pasaia, Gipuzkoa, Spain; arubio@azti.es (A.R.); acaballero@azti.es (A.C.)
2  Centre Oceanogràfic de Balears, Instituto Español de Oceanografía, Lugar Moll Poniente, s/n, 07015 Palma, Balearic Islands, Spain; gabriel.jorda@ieo.es
3  Helmholtz-Zentrum Geesthacht, Institute of Coastal Research, Max Planck Str. 1, 21501 Geesthacht, Germany; jeff.carpenter@hzg.de (J.C.); lucas.merckelbach@hzg.de (L.M.)
*  Correspondence: aivan_625@hotmail.com

**Abstract:** Coastal mesoscale eddies are important oceanic structures partially responsible for regulating ocean-shelf exchanges. However, their description and characterization are challenging; observations are often too scarce for studying their physical properties and environmental impacts at the required spatio-temporal resolution. Therefore, models and data extrapolation methods are key tools for this purpose. Observations from high-frequency radar, one satellite and two gliders, are used here to better characterize the three-dimensional structure of a coastal mode-water eddy from a multiplatform approach in the southeastern Bay of Biscay in spring 2018. After the joint analysis of the observations, a three-dimensional data reconstruction method is applied to reconstruct the eddy current velocity field and estimate the associated water volume transport. The target eddy is detected by surface observations (high-frequency radar and satellite) for two weeks and presents similar dimensions and lifetimes as other eddies studied previously in the same location. However, this is the first time that the water column properties are also observed for this region, which depicts a mode-water eddy behavior, i.e., an uplift of the isopycnals in the near-surface and a downlift deeper in the water column. The reconstructed upper water column (1–100 m) eddy dynamics agree with the geostrophic dynamics observed by one of the gliders and result in cross-shelf inshore (offshore) volume transports between 0.04 (−0.01) and 0.15 (−0.11) Sv. The multiplatform data approach and the data reconstruction method are here highlighted as useful tools to characterize and three-dimensionally reconstruct coastal mesoscale processes in coastal areas.

**Keywords:** eddy; multiplatform observations; 3D data reconstruction; cross-shelf transport; glider; high-frequency radar; mesoscale processes; Bay of Biscay

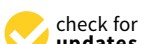

## 1. Introduction

In recent years, several studies have investigated the impact of coastal eddies on ocean-shelf exchanges (e.g., [1–6]), coastal water retention [5,7], as well as the eddy contribution to nutrient and phytoplankton transport in coastal areas (e.g., [1,8,9]) as a key element for primary production. These studies show that knowledge of the coastal eddy field is crucial for a good understanding of physical and ecological processes in the coastal ocean. However, the characterization and prediction of such eddies are difficult due to their interactions with topography and coastal dynamics, which exhibit turbulent and chaotic currents and waves of various sorts.

Huthnance et al. [10] highlighted the importance of numerical models as a methodological tool to extrapolate sparse observations to three-dimensional (3D) fields for use in determining ocean-shelf exchanges in western European shelf seas. At the same time,

numerous studies use models that reproduce coastal eddies to assess the coastal water ocean-shelf exchanges (e.g., [3,6,9,11,12]). In this context, observations are crucial to support numerical models via data assimilation, the validation of results, or as benchmarks to overcome the uncertainties from which models suffer. In addition, observations can be used to simply inter-/extrapolate information in areas of interest.

This study is focused on the southeastern Bay of Biscay (SE-BoB) corner, which is characterized by the presence of canyons (e.g., Capbreton canyon), an abrupt change in the orientation of the coast, and a narrow shelf (Figure 1). The slope current is the main driver of the circulation in the region that in winter flows in the upper 300 m of the water column, advecting warm surface waters eastwards along the Spanish coast and northwards along the French coast (as shown by the solid arrows in Figure 1a), and with a reversed flow in summer that is weaker and less persistent [13–15]. Several studies have investigated open water mesoscale eddies in the southern part of the Bay of Biscay, also called SWODDIES (Slope Water Oceanic eDDIES; e.g., [16–19]), which are observed to be generated along the slope by the interaction of the winter slope circulation with the abrupt changes in bathymetry. Some of these eddies display mode-water eddy behavior, i.e., an uplift of the isopycnals in the near-surface and a downlift deeper in the water column [20]. However, the vertical structure of coastal SWODDIES in the SE-BoB corner has not been well studied. In the comprehensive study of [5], the surface signatures of several coastal anticyclonic SWODDIES were observed in the study area centered around 43.8°N–2.5°W. These SWODDIES originated mainly in winter after the relaxation of strong winter slope current events and had diameters of ≈40–60 km and lifetimes of ≈1–5 weeks. Rubio et al. [5] studied the eddy-induced surface cross-shelf water transport in the study area, concluding that coastal eddies might effectively induce offshore export of coastal waters, as well as their retention in the area. Nevertheless, their analysis was limited to the surface layer, although they suggested the potential importance of a 3D characterization of the transports.

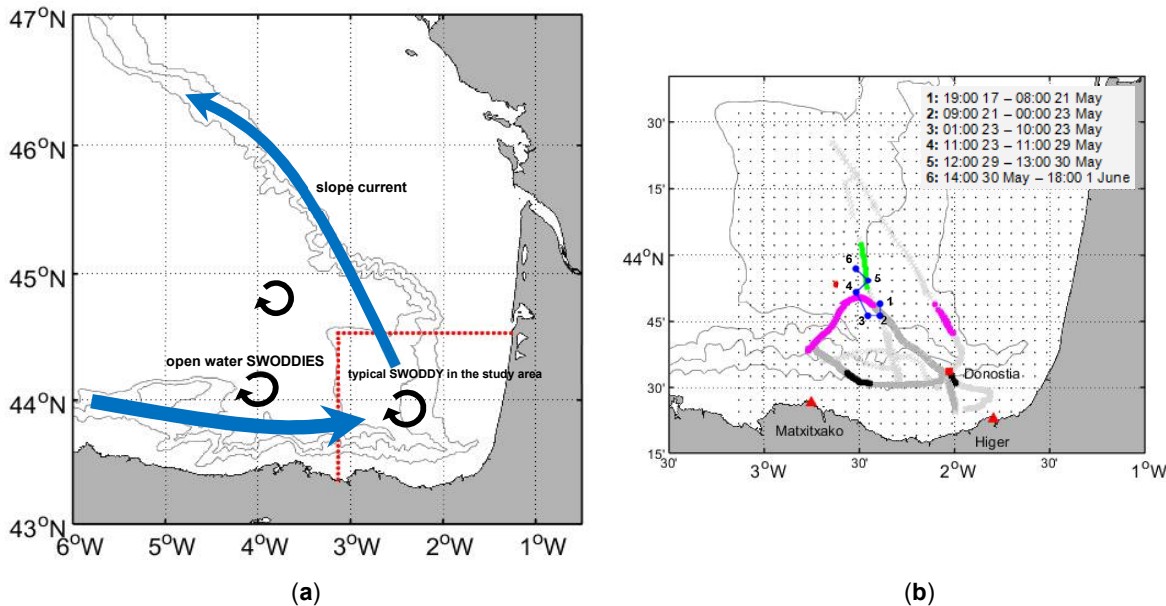

**Figure 1.** (**a**) Location of the study area (dashed red square). The winter slope current is represented by blue solid arrows, whereas the black arrows depict the usual location of anticyclonic eddies. (**b**) Close-up map of the study area. The grid points used to compute total high-frequency radar (HFR) currents are shown by gray dots, and the red triangles show the location of the HFR stations. The red square provides the location of the Donostia mooring. The dark and bright gray crosses show the shallow and deep gliders' trajectories, respectively; while the black, magenta, and green ones show the position of the gliders during the three reconstruction periods (P1, P2, and P3 respectively). The blue dots and numbers show the positions of the anticyclonic eddy core (from the location of the maximum vorticity values). The point used for extracting the wind time series is depicted by the red point. The gray lines show the 200, 1000, and 2000 m isobaths.

In this work, we study a coastal SWODDY from a 3D perspective for the first time in the study area. This SWODDY, analyzed using multiplatform data, is located over the slope and shows a mode-water eddy behavior. This work has a three-fold objective: (i) to characterize the 3D properties of the coastal mode-water eddy, since only the surface signatures of eddies have been described previously in the study area; (ii) to use and assess the skills of a 3D data reconstruction method to reconstruct these kinds of structures and to quantify the cross-shelf transport in the water column; and (iii) to showcase the use of multiplatform multivariable observations for an integrated study of mesoscale processes in a coastal region.

To this end, the data from a high-frequency radar (HFR) system and satellite observations are used to describe the surface signature of the eddy, whereas glider data provide information in the water column at the eddy's periphery and core. HFR and glider data are also used together with a slope mooring and a realistic numerical simulation to obtain 3D reconstructed fields by means of the reduced order optimal interpolation (ROOI) method in three selected periods. The skills of the ROOI method were already analyzed in [21] for this study area.

## 2. Materials and Methods

### 2.1. Multiplatform and Multivariable Data Approach

Several datasets were used in this study to support the analysis of the mode-water eddy detected (objective (i)). Specifically, 2 gliders, a two-station HFR system, the output from the Weather Research and Forecasting (WRF) model and Sentinel-3A satellite data were used for this purpose.

Moreover, HFR and glider data were also used for reconstructing the eddy 3D current velocity fields along with numerical simulation and mooring data (objective (ii)). Details of the different observing platforms and datasets used are summarized in Table 1.

**Table 1.** Summary of the observing platforms and datasets where $\theta$ denotes potential temperature, $\sigma_\theta$ denotes potential density anomaly, Vgeos denotes geostrophic current velocity, U (V) denotes the zonal (meridional) current velocity component, LP denotes low pass filtered and x, y, and z denote the zonal, meridional, and vertical directions, respectively.

| Observing Platforms/ Dataset | Purpose | Variable | Spatial Resolution | Temporal Resolution |
|---|---|---|---|---|
| Glider | Eddy analysis Reconstruction Validation | $\theta$,. Salinity, $\sigma_\theta$, Chl-a, Vgeos $\sigma_\theta$ Vgeos | 3D grid z: 1m x, y: uneven | Uneven Uneven (24 h LP) Uneven |
| HFR | Eddy analysis Reconstruction | U, V | 2D grid x, y: 5 km | Daily + Hourly (10d LP) Average |
| Mooring | Reconstruction | U, V, $\sigma_\theta$ | 1D profile z: 8 m (uneven for $\sigma_\theta$) | Average |
| Sentinel-3A (satellite) | Eddy analysis | Chl-a | 2D grid x, y: 1 km | Daily |
| WRF (model) | Eddy analysis | Wind U, V | Pointwise | Hourly |
| IBI (model) | Reconstruction | U, V, $\sigma_\theta$ | 3D grid x, y: 0.083° z: uneven | Daily |

#### 2.1.1. Temperature, Salinity, Pressure and Chlorophyll-a from Glider Profiles

Temperature, salinity, pressure and chlorophyll-a (Chl-a) data were recovered from two Teledyne Webb Slocum Electric G2 gliders' Conductivity Temperature Depth devices (CTDs, Seabird SBE41 at 1 Hz) and fluorescence sensors (Wetlabs FLNTU at 1 Hz) and used to study the vertical characteristics of the eddy. From these data, potential temperature ($\theta$) and potential density anomaly ($\sigma_\theta$) referenced to the surface and across-track geostrophic current velocities (Vgeos) were also estimated for characterizing the eddy; whilst $\sigma_\theta$ was additionally used as input for the 3D reconstruction of the eddy current velocity fields (for

which it was 24-h low pass filtered). Thanks to the differences between the glider position fixes (measured with GPS) along their trajectories, the average water column current velocities or integrated current velocities were obtained, which were used as reference to estimate the Vgeos.

The gliders were deployed in the region (see Figure 1b) from 16 May until 14 June 2018, during the BB-Trans glider mission run by AZTI (Pasaia, Spain) in collaboration with the Helmholtz–Zentrum Geesthacht (HZG) (Geesthacht, Germany) in the frame of the JERICO-Next European project Transnational Actions (https://www.jerico-ri.eu/ta/select ed-projects/second-call/bb-trans/ (accessed on 9 December 2020)). During this mission, a shallow-water glider (0–100 m depth, hereinafter shallow glider) and a deep-water glider (0–1000 m depth, hereinafter deep glider) were deployed. The shallow glider comprises data from 16 May to 29 May, while the deep glider comprises data from 17 May to 14 June. After the glider mission, the data were processed, quality controlled, and made freely available by the EGO project and the national programs that contribute to it. The data flagged as bad data were removed, as well as the data corresponding to excessive or low glider vertical velocities, setting a few gaps within the glider's datasets. In addition, transects were linearly interpolated in the water column to the mean latitude and longitude (of each transect), obtaining vertical profiles with a vertical resolution of 1 m.

### 2.1.2. Surface Currents from HFR

Hourly fields of surface current velocities from the Basque Operational Oceanography System (EuskOOS, https://www.euskoos.eus/en/ (accessed on 9 December 2020)) HFR (CODAR Seasonde) were used both for studying the surface characteristics of the detected eddy and as input for the 3D reconstruction of the eddy current velocity fields. The current velocity data were quality controlled using procedures based on velocity and variance thresholds, signal-to-noise ratios, and radial total coverage, following standard recommendations [22]. To isolate the most persistent signals in the HFR surface current velocity fields, the 10-day low pass filtered fields (hereinafter LP fields) were computed using an 8th order Butterworth filter. The daily averages were also used for characterizing the eddy at 3 days. Moreover, HFR data were averaged to be used as input for the 3D reconstruction.

The EuskOOS HFR system consists of two sites, one in Cape Higher and another one in Cape Matxitxako (see Figure 1b). It works at a central frequency of 4.46 MHz with an operacional bandwidth of 30 kHz. The footprint area covers ≈150 km off the coast and the integration depth is ≈1.5 m. The system has provided hourly current velocity fields gridded onto a 5 km resolution regular orthogonal mesh since 2009, with some interruptions mostly due to maintenance stops or malfunctioning related to severe atmospheric conditions. The performance of this system and its potential for the study of ocean processes and transport patterns have already been demonstrated by previous works (e.g., [5,15,23–25]).

### 2.1.3. Vertical Profiles of Potential Density Anomaly and Currents from a Moored Buoy

Current velocities from the acoustic Doppler current profiler (ADCP) and $\sigma_\theta$ computed from the conductivity temperature devices (CTs) along the first 100 m of the water column at the Donostia slope mooring were used only for the 3D reconstruction of the eddy current velocity fields. To that end, the ADCP data were quality controlled by beam amplitude and correlation magnitudes and velocity errors following [26]. In addition, as with the glider and HFR data, mooring data were adapted (averaged) to be used as input for the 3D reconstruction.

The EuskOOS Donostia mooring (Wavescan Buoy WS169) is anchored in a water depth of 550 m on the Spanish slope (at 43.56°N–2.03°W, see Figure 1b) and has been providing data since 2007. Among the oceanographic sensors, a downward-looking ADCP (RDI Workhorse), operating at a frequency of 150 kHz, measures current velocities unevenly distributed in time (with a mean temporal spacing of about an hour) with bins of 8 m depth starting at a depth of 12.26 m and extending down 200 m into the water column. In addition,



several CTs (Seabird SBE37IM at hourly measurements) along the mooring line provide hourly temperature and salinity data at $-10$, $-20$, $-30$, $-50$, $-75$, $-100$ and $-200$ m (note that the CT located at $-200$ m also contains a pressure sensor). The performance of these sensors and the quality of the data have already been demonstrated (e.g., [15,24,27,28]).

### 2.1.4. Chl-a Images and Wind Data

Chl-a images and wind data were used for studying the surface characteristics of the detected eddy. Daily level 3 Chl-a images (1 km resolution) were provided by the Ocean and Land Colour Instrument (OLCI) airborne in the Sentinel-3A satellite during the BB-Trans mission. However, cloud-free images were only obtained for 20, 21, and 24 May due to cloudy conditions during the glider mission period.

The hourly wind data were extracted from the Weather Research and Forecasting model (WRF) provided by the meteorological agency of Galicia (MeteoGalicia) at 43.89°N–2.62°W (Figure 1b). This model, with a native resolution of 12 km, reproduces the offshore wind fields of the SE-BoB with reasonable accuracy [29].

### 2.1.5. Numerical Simulations

As in [21], the covariance matrix needed for the 3D reconstruction using the ROOI method was built upon the IBI_REANALYSIS_PHY_005_002 product (hereinafter IBI), provided by the Copernicus Marine Environment Monitoring Service for the period 1992–2016. The IBI reanalysis is based on a realistic configuration of the NEMO model for the Iberian Biscay Irish region, which assimilates in situ and satellite data. For more details on the simulations, the reader is referred to [21] (Table 1). A complete description of the product and its validation can be found in [30] and the following links: http://cmems-resources.cls.fr/documents/PUM/CMEMS-IBI-PUM-005-002.pdf (accessed on 9 December 2020) and http://resources.marine.copernicus.eu/documents/QUID/CMEMS-IBI-QUID-005-002.pdf (accessed on 9 December 2020).

The temporal resolution of the dataset used was daily, and the horizontal spatial resolution was $0.083° \times 0.083°$ ($\approx$6–9 km). The vertical levels used were unevenly distributed with separations between 1 and 3 m in the first meters and an increasing separation with depth.

### 2.2. *Method for the 3D Reconstruction of the Observed Fields*

In this section, the 3D data reconstruction method and its implementation in our specific case are explained. Note that to allow the blending of datasets with different temporal and spatial resolution, a specific preprocessing was needed as explained in Section 2.2.2.

### 2.2.1. The ROOI Method

The ROOI method was used to reconstruct the 3D current velocity fields from HFR surface fields, mooring, and glider observations. In [21], after reviewing several methods to expand the HFR data to subsurface layers, two data reconstruction methods, namely the discrete cosine transform-penalized least square (DCT-PLS) [31] and the ROOI [32], were tested and compared in the study area. The ROOI provided better results in areas far from the observations, thus being more suitable for the present study, since it allowed taking advantage of the complementarity of the available datasets. More quantitatively, the ROOI provided mean spatial reconstruction errors between 0.55 and 7 cm/s and mean relative errors of 0.07–1.2 times the root mean square value for the first 150 m depth. The ROOI is based on Empirical Orthogonal Function (EOF) decomposition and was first proposed by [32] to reconstruct sea surface temperatures. Since then, it has been used to reconstruct several variables such as sea level pressure [33], sea level anomalies [34], or 3D current velocity fields [35]. The details of the method can be found in [32,33,35], so here, only the basics are presented to understand the method and support the discussion of the results.

The 3D current velocity field we want to obtain can be expressed as a $m \times n$ matrix $Z(r, t)$, where $r$ is the $m$-vector of spatial locations and $t$ is the $n$-vector of times. However, the observations only cover a small part of the 3D domain (a few $N$ locations, being $N \ll m$). In order to obtain current velocities in all $m$ locations, an EOF decomposition approach is considered. If a spatial covariance matrix is computed as $C = n^{-1}ZZ^T$, an EOF decomposition can be applied:

$$C = U\Lambda U^T \tag{1}$$

where $U$ is an $m \times m$ matrix whose columns are the spatial modes (EOFs) and $\Lambda$ is the $m \times m$ diagonal matrix of eigenvalues. Then, the velocity field can be exactly reproduced as:

$$Z(r, t) = U(r) \cdot \alpha(t) \tag{2}$$

in which the temporal amplitude $\alpha$ can be computed as $\alpha = U^T Z$, since $U$ is orthogonal.

Since there are not enough observations for directly computing $Z$, historical data from a realistic numerical simulation are used to represent the current velocity statistics ($C$) between all $m$ locations. Then, $U$ is inferred applying the EOF decomposition. In addition, Equation (2) is truncated to include only the $M$ leading EOFs that reproduce the features that we intend to reconstruct, avoiding introducing noise from the higher order modes. Thus:

$$Z_M(r, t) = U_M(r) \cdot \alpha_M(t) \tag{3}$$

Then, the $\alpha_M$ can be determined under the constraint that the reconstructed $Z_M$ fits the observations available at each time step, while minimizing a cost function that takes into account the observational error and the role of neglected modes (see [32,33] for the complete derivation). In summary, the values of the current velocities of a 3D grid can be obtained merging (i) the spatial modes of variability computed from a realistic numerical simulation, and (ii) the temporal amplitudes obtained using the available observations.

Note that the $m$-vector of spatial locations (each row of $Z$) corresponds to the grid points at which we want to reconstruct the currents as well as the grid points where we have the observations. Each grid point only corresponds to one velocity component. Thus, if instead of using only one component both horizontal components are jointly used, there will be twice the number of grid points. However, more variables can also be used, and the way to incorporate them is simply enlarging the matrix $Z$ with additional rows: one for each grid point at which the new variable is available.

### 2.2.2. Implementation of the ROOI Method

Current velocity (zonal (U) and meridional (V)) and $\sigma_\theta$ observations were used to reconstruct current velocities at the IBI grid points (see Section 2.1.5), while the ROOI also reconstructed the observations at their own locations (fitting the reconstruction to the observation). For that, first, spatial covariances were set from IBI between current velocities at the IBI grid points and current velocities and $\sigma_\theta$ at the location of the observations (interpolated from the IBI grid). Then, based on these covariances and the observations, the reconstructions were obtained following the methodology explained in Section 2.2.1.

Current velocity fields were reconstructed for three different periods (hereinafter called P1, P2, and P3), as indicated in Table 2. The criteria for selecting the periods was to have the eddy or its near field measured simultaneously by at least the HFR and one glider. In P1 and P2, the eddy was detected at the surface by the HFR, while the gliders surveyed a frontal area to the south of the eddy (P1) and the eddy periphery (P2). In P3, the HFR detected a weak anticyclonic signal at the surface, while the deep glider crossed the eddy core. The relative positions of the gliders with the overall locations of the eddy cores for each period can be seen in Figure 1b; more accurate relative positions are provided in Figure 2a for P1, Figure 2b for P2, and Figure 2c for P3. P3, in addition to the reconstruction, is used to study the hydrographic vertical properties of the eddy, since it is the period when the deep glider crossed the eddy core.

**Table 2.** Definition of the three periods in 2018 used for the reconstruction, with all times in UTC. Period P3 was also used for the vertical description of the eddy.

|    | **From** | **To** |
| --- | --- | --- |
| P1 | 17:00 20 May | 17:00 21 May |
| P2 | 00:00 25 May | 23:59 26 May |
| P3 | 02:37 2 June | 02:06 3 June |

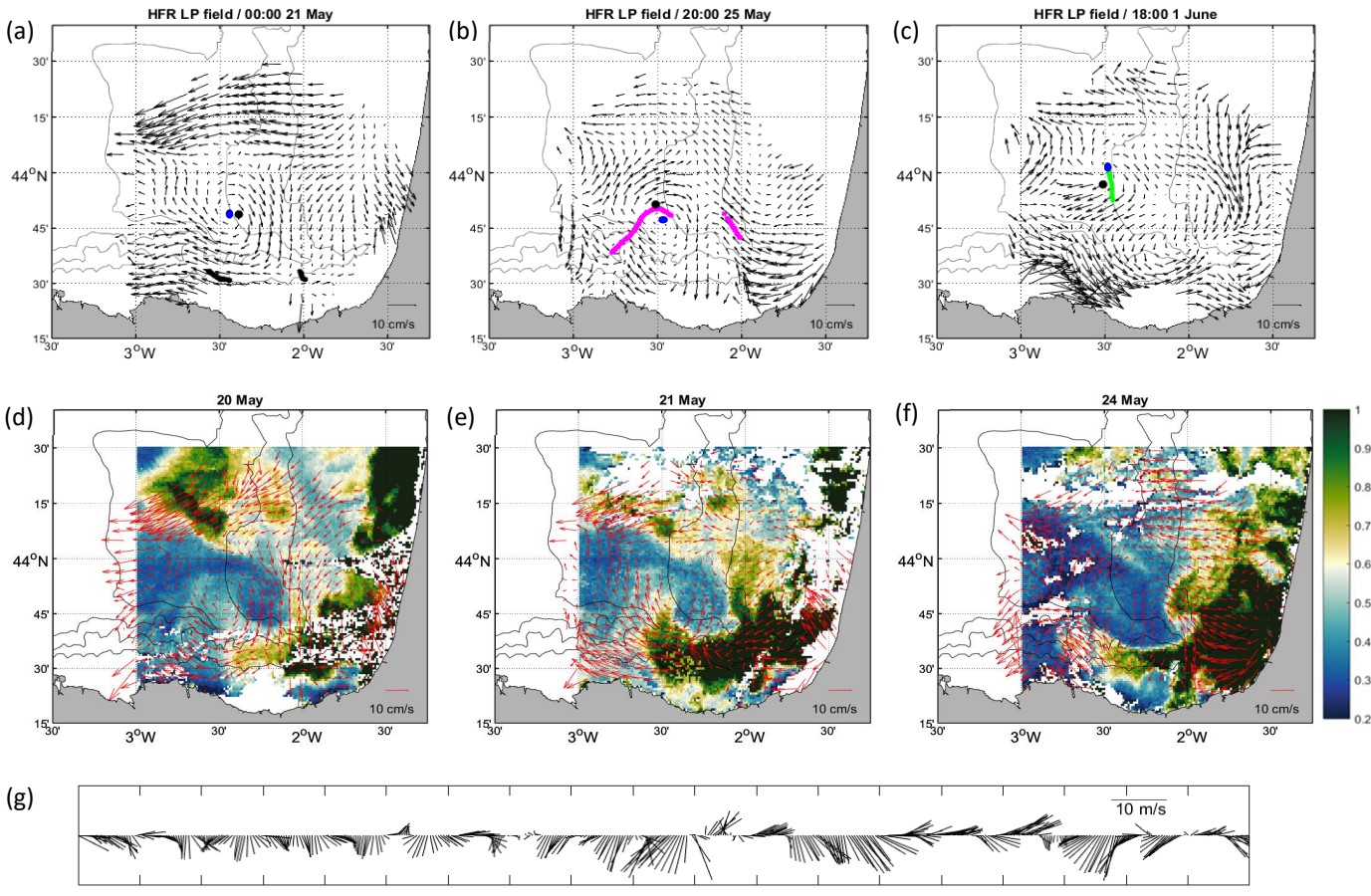

**Figure 2.** In (**a–c**) the black dots show the positions of the eddy core, estimated from the location of the maximum of relative vorticity, overlaid to a snapshot of the high-frequency radar (HFR) low pass filtered (LP) field for a date within P1 (**a**), P2 (**b**), and the last detection date before P3 (**c**). In (**a–c**), the trajectory followed by the gliders is shown in black, magenta, and green colors, respectively, and the blue dots depict the position of the eddy core. (**d–f**) show satellite Chl-a (mg/m$^3$) images with the daily mean HFR fields superimposed (red arrows) showing the mentioned anticyclone. The gray lines show the 200, 1000, and 2000 m isobaths. (**g**) depicts the wind series (see location in Figure 1b).

Note that apart from current velocity data (from HFR and ADCP), $\sigma_\theta$ from the mooring and glider CTDs was also used as an input variable for the 3D reconstruction of the current velocity field. $\sigma_\theta$ was selected because it presented significant covariances with the current velocities and gathered the effect of salinity and $\theta$. Note that another variable that presented significant covariances was the across-track Vgeos; however, it was not used as an input variable but as reference to validate the results.

Considering that the historical (hindcast) data from IBI were provided on a daily basis and the covariances could not resolve features corresponding to shorter temporal scales, all the observations were averaged or low-pass filtered and subsampled to a daily scale, accordingly. HFR and mooring CT and ADCP data were averaged for each reconstruction period, making a daily average for P1 and P3, and a 2-day average for P2. Then,

glider data were not averaged but 24-h low pass filtered to avoid aliasing of signals, due to the continuous change in position of the glider throughout the time. Note that the adapted observations used for the reconstruction in each of the three periods are shown in Supplementary Material 1 (SM1, Figure S1).

Several sensitivity tests were performed to tune the method's parameters. The main parameters to be adjusted were M, the number of modes, and the error corresponding to velocity and $\sigma_\theta$ observations ($\varepsilon_{vel}^{obs}$ and $\varepsilon_{pd}^{obs}$, respectively). After several tests, M = 1000 modes were used, since they provided the necessary variability to describe the structures that were observed during each reconstruction period. Regarding the observational errors ($\varepsilon^{obs}$), the weight given to an input observation in the reconstruction can be partially increased (decreased) by decreasing (increasing) the $\varepsilon^{obs}$ associated with it. Therefore, a different choice was made for each period, since different features were detected by each observing platform at each time, and a different weight was given to each variable. Thus, for P1, $\varepsilon_{vel}^{obs}$ = 2 cm/s and $\varepsilon_{pd}^{obs}$ = 0.4 kg/m$^3$, whereas for P2 and P3, $\varepsilon_{vel}^{obs}$ = 2 cm/s and $\varepsilon_{pd}^{obs}$ = 0.1 kg/m$^3$. However, it is worth mentioning that the sensitivity of the results to different parameterizations is relatively low, as long as the used $\varepsilon^{obs}$ values are reasonable for each observing platform. A high $\varepsilon^{obs}$ provides results that appeared overly smoothed, whereas a very low $\varepsilon^{obs}$ provides noisy results (the sensitivity tests carried out with a set of different values for these two parameters are shown in SM2, Figures S2–S14).

Since some observations were limited to the upper 100 m (i.e., the shallow glider observations) and [21] showed that the skills of the 3D reconstruction decreased with depth, the reconstructions were made for the upper 100 m, and only using observations within that range. Moreover, the deep glider observations under −100 m were used to validate the results.

### 3. Results

*3.1. Observed 3D Properties of the Eddy*

The HFR LP fields were used to study the persistent (≈15 days) surface signature of the eddy. The location of the anticyclonic eddy core was tracked in the HFR LP fields by the position of the maximum relative vorticity (Figure 1b). It was first observed on 17 May, and its core moved gradually northwestward by 18 km until 1 June. The eddy had a quicker northward displacement (first northeastward and after northwestward) in the last three days when weaker winds from the north were observed (Figure 2g). Additionally, Chl-a images show that the Chl-a distribution patterns in the nearby areas of the eddy on 20, 21, and 24 May (Figure 2d–f) are in agreement with the daily mean HFR field patterns that are superimposed (red arrows) and with the eddy core locations.

Figure 2a–c provides three snapshots of three different moments of the surface evolution of the eddy, showing the position of the eddy cores overlaid to the HFR LP fields and the relative position of the gliders. These three moments correspond to a date within P1, a date within P2, and the last detection date at the surface (the day before P3), respectively. In these snapshots, it is observed that the eddy core locations provided by the maximum relative vorticity (black dot) are approximate; sometimes, they are accurate, as in Figure 2a, where the HFR LP field shows that the eddy core (blue dot, determined visually) is almost in the same location as the maximum relative vorticity. Some other times, they are less accurate, as in Figure 2b,c, where the maximum vorticity is not precisely on the center of the eddy shown by the HFR LP fields. In any case, the obtained locations with the relative vorticity maximum are accurate enough to track the overall trajectory of the eddy. Note that in Figure 2b, it can be observed how the shallow glider passes along the periphery of the eddy in P2. On the other hand, Figure 2c shows that the eddy core is detected by the HFR LP fields on 1 June at 44.02°N–2.48°W, where the deep glider passes through on 2 June (green crosses), i.e., in P3, when the surface signature shows a weak anticyclonic flow (as shown in Figure S1h in SM1). This indicates that the mode-water eddy observed by the deep glider in P3 is the same as that observed at the surface until 1 June.

As explained before, the hydrographic properties of the eddy have been analyzed for P3, when the deep glider crossed the eddy core. The θ and σ_θ profiles (Figure 3b,c) show that at shallow depths, the isotherms and isopycnals are uplifted between −40 and −200 m. In the case of the salinity (Figure 3a), the uplifting starts at −100 m, whereas in the Chl-a profile (Figure 3d), it starts at around −70 m. The seasonal thermocline is located between −10 and −20 m without any remarkable uplift. The across-track Vgeos retrieved from the hydrography of the upper 100 m (i.e., referenced to a level of no motion at −100 m) clearly shows a cyclonic behavior (Figure 4a). Note that this reference level is only considered to show the dynamics provided by the waters within the first 100 m. We will show that for the whole Vgeos construction, the whole water column should be considered.

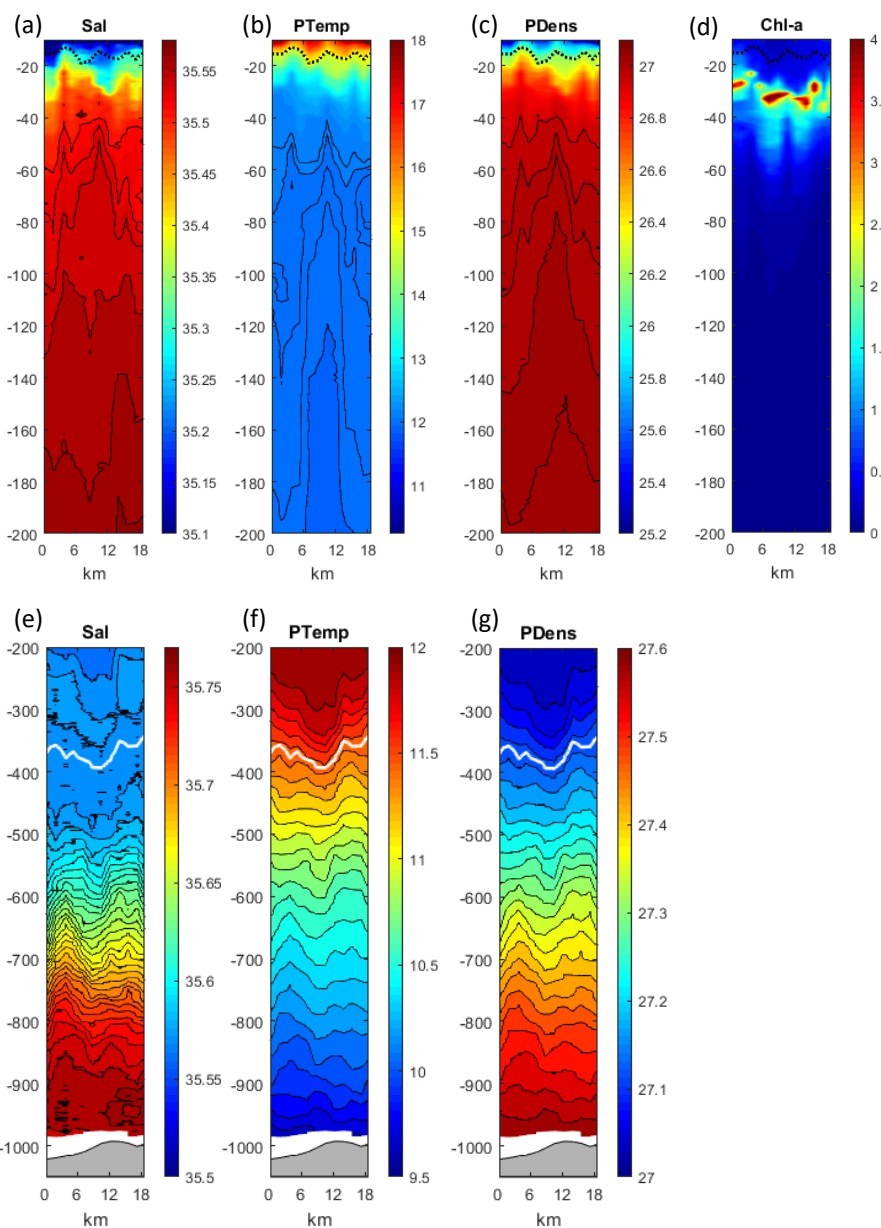

**Figure 3.** Deep glider (**a**,**e**) salinity (psu), (**b**,**f**) θ (°C), (**c**,**g**) σ_θ (kg/m³), and (**d**) Chl-a (mg/m³). From −10 to −200 m for (**a**–**d**); and from −200 m to the bottom (gray area) for ((**e**–**g**), note the change in the color bar). In (**a**–**d**), the dashed black line depicts the 15 °C isotherm (representing the form of the seasonal thermocline), whereas in (**e**–**g**), the white line depicts the 11.5 °C isotherm (representing the form of the permanent thermocline). The *X*-axis shows the distance (in km) to the first point of the profile (from north to south in the maps shown in Figures 1b and 2c) and the *Y*-axis the depth in m.

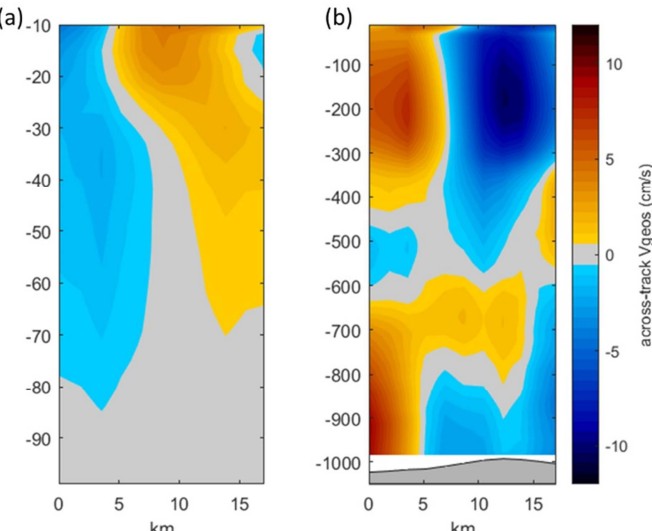

**Figure 4.** Across-track Vgeos (cm/s) profiles corresponding to P3. The Vgeos was referenced to a level of no motion at −100 m for (**a**) and the integrated currents for (**b**). The positive values correspond to the eastward across-track currents, whereas the negative ones correspond to the westward currents. The *X*-axis shows the distance (in km) to the first point of the profile (from north to south in the maps shown in Figures 1b and 2c) and the *Y*-axis the depth in m. In (**b**) the bottom is depicted by dark gray color.

In contrast to this uplift observed in shallow depths, in deeper waters, a downlift is observed, showing a structure typical of mode-water eddies. The isotherms and isopycnals downlift approximately from −200 m to −450 m (Figure 3f,g), whereas in the case of the isohalines, the downlift starts at −100 m (Figure 3e). The downlift is shown by a minimum at the core of the eddy, which is deeper than the isoline on the sides of the profile. The 11.5 °C isotherm gives an idea of the shape of the permanent thermocline, which in Figure 3e–g shows a downlift at around −400 m and thus shows an anticyclonic signal. In addition, the deep salinity profile (Figure 3e) shows a salty water lens with its core centered at around −250 m and its base at around −350 m, which is downlifted by the anticyclone. The across-track Vgeos also shows that the anticyclonic signal extends until around −450 m (Figure 4b). Note that the Vgeos is referenced to the integrated currents, thus considering the water properties of the whole water column.

A rough theoretical estimation of the depth of an eddy can be obtained by the factor W·f/N (e.g., [36,37]), where W is the width (diameter of the eddy), f is the Coriolis frequency, and N is the buoyancy frequency. Based on the measurements taken during P3, when the eddy core is sampled, we find $W \approx 16$ km (Figure 4b) and $N \approx 3.4 \cdot 10^{-3}$ 1/s beneath the thermocline, leading to the rough depth estimate of −480 m. On the other hand, based on the azimuthal velocity Vr = 9 cm/s and the radius of the eddy R = 8 km, the obtained Rossby number value, defined as Ro ≡ 2 Vr/(R·f), is 0.22.

### 3.2. Reconstruction of the 3D Eddy Current Velocity Fields

In this section, we focus on the reconstruction of the detected eddy; however, the results over the whole study area are depicted to show the overall variability in the reconstructed fields.

### 3.2.1. Skill of the Reconstruction

Since the skill of the 3D reconstructions cannot be validated with additional external observations of current velocities, one option is to compare the reconstructed current velocity fields with the observations (previously used as inputs) themselves (as in [35]). Thus, the root mean square differences (RMSDs) between the reconstructed and observed current velocities at the HFR and ADCP observation points were estimated (shown in Table 3). Note that the points considered correspond to the whole grid and not necessarily

to locations where the eddy is observed. The RMSD at the HFR observation points show values ranging from 1.18 to 1.74 cm/s, while at the ADCP observation points, RMSD values range from 0.58 to 1.74 cm/s. The root mean square values range from 3.45 to 5.6 cm/s and from 0.70 to 3.19 cm/s at HFR and ADCP observation points respectively, showing that the obtained RMSD values are low.

**Table 3.** Root mean square differences (RMSDs) between the reconstructed and observed current velocities at the HFR and acoustic Doppler current profiler (ADCP) observation points (in cm/s).

|    | $U_{HFR}$ | $V_{HFR}$ | $U_{ADCP}$ | $V_{ADCP}$ |
|----|-----------|-----------|------------|------------|
| P1 | 1.39      | 1.37      | 0.58       | 0.68       |
| P2 | 1.33      | 1.18      | 1.74       | 0.58       |
| P3 | 1.72      | 1.74      | 1.35       | 0.92       |

### 3.2.2. Reconstructed 3D Eddy Current Velocity Fields and Associated Transports

In P1, the 3D reconstruction of the eddy current velocity fields (Figure 5) shows that the anticyclone is centered at 43.75°N–2.33°W in the first 50 m, whereas from −50 to −100 m, it is slightly displaced northwestward and horizontally spread. The eddy signal is weakened with depth and shows progressively weaker relative vorticity. In Figure 5d, the reconstructed currents projected across the section shown by the black line in Figure 5a–c (hereinafter section A) also show a clear anticyclonic behavior. The across-track currents are stronger in the first 30 m, thus providing higher anticyclonic (i.e., negative) relative vorticity values.

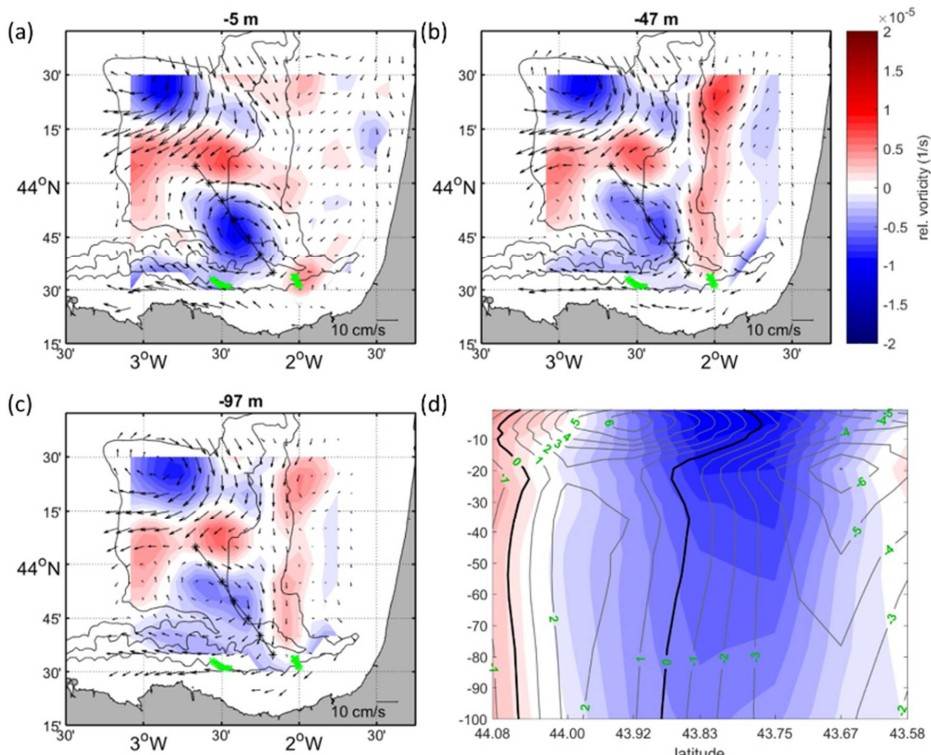

**Figure 5.** Reconstructed fields in P1. (**a**–**c**) show the current velocity fields and the relative vorticity for three depth levels. The gray lines show the 200, 1000, and 2000 m isobaths. The green crosses show the position of the gliders and mooring observations. The straight black line depicts section A. In (**d**), the relative vorticity in section A is shown along with the velocities perpendicular to it (gray contour lines). The values of the velocities are depicted in green in cm/s, and the 0 cm/s contour is marked in black. The positive/negative velocities correspond to northeastward/southwestward currents. The *Y*-axis shows the depth in m.

In P2 (Figure 6), the signal is narrower and more intense than in P1 (Figure 5). From the surface to −15 m, the eddy is centered at around 43.83°N–2.5°W; however, its core is displaced southeastwards between −15 and −25 m to 43.75°N–2.33°W. This latter position corresponds to the position of the eddy in P1, suggesting a tilting of the eddy, where the first 15 m have been moved northwestward, while the deeper part of the eddy remains in its original place (as shown in Figure 6d). The across-track current velocity contours of section A show a clear anticyclonic flow that is weaker at the surface (where the eddy is tilted) and is stronger for subsurface levels.

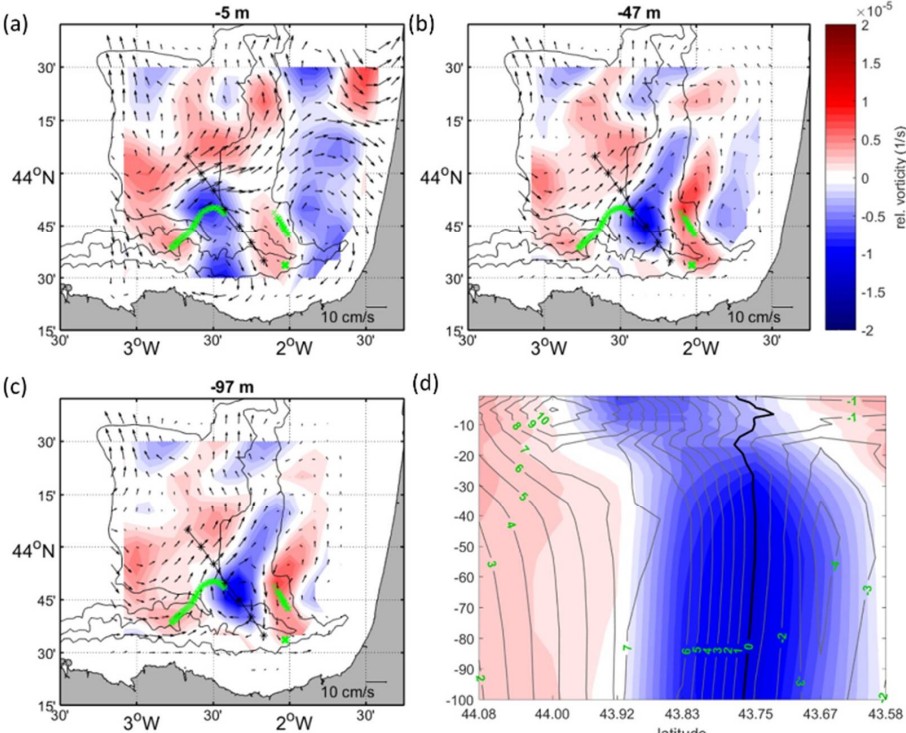

**Figure 6.** Reconstructed fields in P2. (**a–c**) show the current velocity fields and the relative vorticity for three depth levels. The gray lines show the 200, 1000, and 2000 m isobaths. The green crosses show the position of the gliders and mooring observations. The straight black line depicts section A. In (**d**), the relative vorticity in section A is shown along with the velocities perpendicular to it (gray contour lines). The values of the velocities are depicted in green in cm/s, and the 0 cm/s contour is marked in black. The positive/negative velocities correspond to northeastward/southwestward currents. The *Y*-axis shows the depth in m.

In P3 (Figure 7), the weak anticyclonic signal detected at the surface over the trajectory of the deep glider is expanded for the whole water column without any horizontal tilting. Its relative vorticity is the lowest between −20 and −30 m, while it increases at around −10 and −60 m. The across-track current velocity contours with respect to section A also show an anticyclonic flow. The observed across-track Vgeos and the reconstructed currents projected across the trajectory of the deep glider show a similar anticyclonic pattern at −50 and −90 m (Figure 8) with eastward currents in the northern part of the trajectory and westward currents in the southern part.

The diameter of the reconstructed eddy (along the section A and taking as a reference the contour of negative relative vorticity) in the three periods considered is around 30 km, 25 km, and 15 km, respectively for P1, P2, and P3. The transport induced by the reconstructed eddy (delimited again by the negative vorticity) was also estimated at each period for the upper 100 m across section A. The positive transports correspond to the northeastward across-track direction (inshore), whereas the negative ones correspond to the southwestward one (offshore). Although this section is not strictly parallel to the shelf,

it crosses the cores of the three reconstructed eddies, being useful to provide a rough estimate of cross-shelf transports across the same section. The estimated positive (negative) values are 0.05 Sv (−0.11 Sv), 0.12 Sv (−0.05 Sv), and 0.15 Sv (−0.01 Sv) in P1, P2, and P3, respectively.

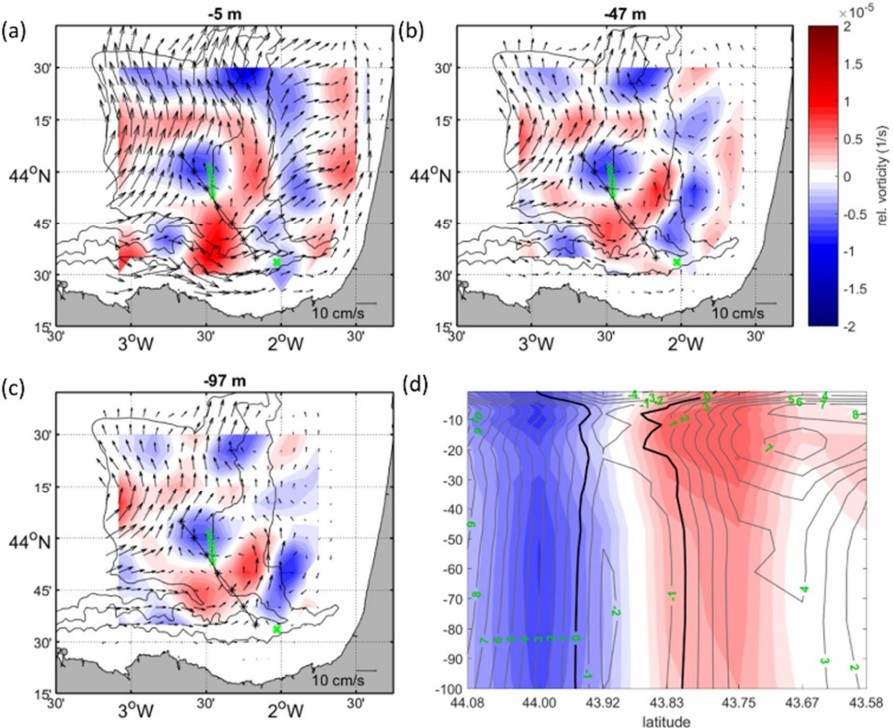

**Figure 7.** Reconstructed fields in P3. (**a–c**) show the current velocity fields and the relative vorticity for three depth levels. The gray lines show the 200, 1000, and 2000 m isobaths. The green crosses show the position of the gliders and mooring observations. The straight black line depicts section A. In (**d**), the relative vorticity in section A is shown along with the velocities perpendicular to it (gray contour lines). The values of the velocities are depicted in green in cm/s, and the 0 cm/s contour is marked in black. The positive/negative velocities correspond to northeastward/southwestward currents. The *Y*-axis shows the depth in m.

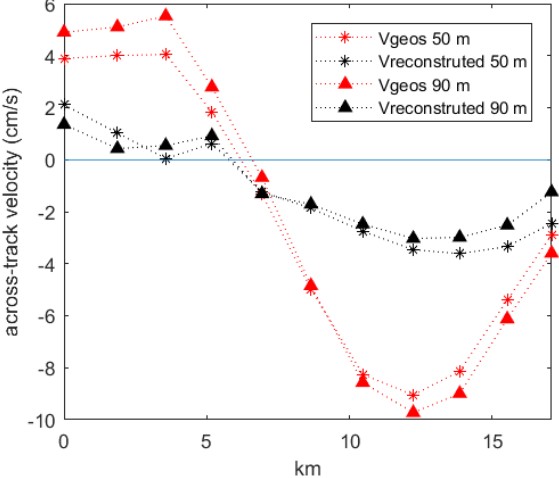

**Figure 8.** Across-track currents (in cm/s) along the deep glider trajectory in P3 for −50 m (asterisk) and 90 m (triangle). The red markers correspond to the Vgeos observed by the glider (as in Figure 4b), whereas the black markers correspond to the reconstructed current velocities. Eastward currents have positive values whereas westward currents have negative values.

## 4. Discussion

Thanks to a multiplatform data approach, combining in situ and remote sensing data, a coastal mode-water eddy was detected and investigated in 3D for the first time in the study area. From the observation of the surface anticyclonic signature of the eddy in the HFR LP fields, we conclude that the structure remained in the area for two weeks. This anticyclonic signature is also coherent with the Vgeos computed along the deep glider's trajectory (Figure 4b). The location and the main surface characteristics of the mode-water eddy observed here (Figures 1b and 2) are in agreement with the characteristics of anticyclonic eddies previously studied at the surface in the area by [5] who suggested that eddies with diameters of $\approx$40–60 km and lifetimes of $\approx$1–5 weeks were recurrent in this region. During the period of observation, the eddy showed a slow drift northward along the French shelf-break following a wind change from northerlies to southwesterlies. This behavior is also coherent with the observations of [5], where at least one of the observed structures (i.e., the anticyclonic eddy A1214 in their paper) followed a very similar path.

With regard to the vertical properties of the mode-water eddy, we observe that the waters within the eddy were uplifted between $-40$ and $-200$ m and downlifted between $-200$ and $-450$ m. In a previous glider mission, an open water mode-water eddy west of the study area was detected by [19]. In that study, the mode-water eddy was located in deeper waters and showed larger horizontal and vertical dimensions, being the uplifting and downlifting of the isotherms and isohalines placed at shallower and deeper depths, respectively. The vertical properties of the mode-water eddy observed here extend until depths of around $-450$ m (Figures 3 and 4). These observations are coherent with the values obtained from the rough theoretical estimation of the depth of the eddy, based on the width and the f/N ratio [36,37], which provides a depth of $-480$ m. In addition, the Ro = 0.22 is similar to values found in [37] and indicate that the eddy is largely in geostrophic balance.

The mode-water eddy observed by [19] showed an anticyclonic flow on the surface by sea level anomaly altimetry maps that was not observed in our case (not shown). This could be related to the closer vicinity of this structure to the coast [38], which would affect the altimetry measurements, since both HFR surface fields and Chl-a images show the existence of an anticyclonic eddy. The weak surface signature of the eddy in the HFR fields in the last days could be related to the partial compensation between the cyclonic (baroclinic) and anticyclonic (barotropic) signals of the mode-water eddy upper layers as suggested by [39]. However, in mode-water eddies, the geostrophic velocities are dominated by the permanent pycnocline, obtaining the same direction of rotation as in anticyclones [40]. In fact, it is shown that the geostrophy provided by the hydrography of the whole water column (Figure 4b) compensates the cyclonic behavior provided in the shallowest levels (Figure 4a). It is also worth mentioning that 3D studies such as this one are key, since mode-water eddies are difficult to track by altimetry and HFR due to their smaller surface signal.

An in-depth discussion on the possible mechanisms behind the generation of this eddy and its mode-water structure is not possible, since it can be barely supported by the available observations. However, an approximate description of the history of the eddy could be the following. Before the first detection of its surface signature in the HFR fields, an event of intensified along self/slope current was detected in the study area between 7 May and 13 May (not shown). After this intensification, an anticyclonic meandering on 15 May preceded the day when the eddy was first detected (17 May). The interaction of the intensified slope current with the bathymetry is one of the main drivers of the generation of SWOODIES in the study area [16,18,41], which was also suggested to be responsible for the generation of the eddy A1214 studied by [5]. However, whether the eddy was already formed during the event of intensified along shelf/slope current remains unclear. Concerning the generation of mode-water eddy in the water column, the observed wind variability in the area might help to develop this kind of structure through the Ekman pumping triggered by the eddy-wind interactions as explained by [39].

A rough estimate of the vertical velocity due to this effect shows values around 18 cm/day, which are insufficient for the generation of the mode-water eddy in the period where the eddy is detected; however, the permanent underlying mean cyclonic circulation of this coastal area might also reinforce this effect. Further work through additional observations and process-oriented numerical experiments should be addressed to better understand the triggers for the generation of mode-water eddies in the study area, in view of their implications on the vertical and cross-shelf transport of high Chl-a to coastal waters [5,19].

In the future, in order to better characterize eddies in the study area, an ad hoc sampling strategy would be desirable. HFR and satellite observations could help to detect them before a survey with in situ data, in order to plan the timing and location of the deployments. For instance, an improved strategy could be based on the deployment in the area covered by the HFR of two deep gliders that sample the eddy core and periphery and also cross the slope and shelf-break areas in perpendicular directions. In this way, the vertical properties and the across-track geostrophic currents would be better characterized and provide information about the cross-shelf and along-shelf water transports in the water column. Current measurements provided by gliders or by additional observing platforms such as shipborne downward-looking ADCPs or drifting buoys would also be valuable to analyze the eddy and to validate the reconstruction (which is independent of the data used, so it can accommodate different sampling strategies as shown in [21,35]). In addition, turbulence measurements in the core and periphery of the eddies would be very valuable to study eddy-induced mixing processes. These kinds of multiplatform sampling strategies set from mid to long-term temporal frames arise as a helpful approach toward the comprehensive characterization of eddies and their effects on coastal ecosystems.

Building on the work by [21], this work demonstrates the potential of a data reconstruction method for retrieving the 3D current velocity field associated with eddies in coastal regions. Concerning the methodology used for the reconstruction, one important factor to consider when applying the ROOI is the required input parameters ($M$, $\varepsilon_{vel}^{obs}$, $\varepsilon_{pd}^{obs}$). Low RMSD values between the reconstructed and observed fields at the observation points (Table 3) and a realistic variability of the reconstructed maps have been the criteria used to select the optimal parameters for the reconstruction (sensitivity tests displayed in SM2, Figures S2–S14). Note that the sensitivity of the results to the different parameterizations is reduced, as long as the number of selected modes provides the minimum variability needed for the reconstruction of the target processes and the observational errors are reasonable.

The reconstructions were carried out for three different periods where the eddy and its surroundings were sampled in a different manner. In P1, the eddy was only observed by the HFR on the surface, whereas in P2, the HFR observed the eddy, and the shallow glider sampled the periphery. In P3, the surface eddy signal was very weak, but the deep glider sampled its core. The ROOI was able to reconstruct the eddy for the three periods, showing its ability to carry out reconstructions under different scenarios.

The ability of the method to reconstruct the eddy is especially remarkable in P3, taking into account its complex vertical structure in terms of $\sigma_\theta$ fields and its weak surface signature. Indeed, the ROOI reproduces the anticyclonic flow of the eddy in the upper 100 m, extending the weak anticyclonic signature observed at the surface, despite the cyclonic pattern provided by the $\sigma_\theta$ observations in the water column (shown in Figure 3c and also noticeable in Figure 4a). This anticyclonic flow in P3 is still reconstructed even giving more weight to $\sigma_\theta$ observations (by tuning the $\varepsilon^{obs}$ as explained in Section 2.2.2; sensitivity tests are shown in Figures S13 and S14 in the SM2) and might be explained due to higher covariances between surface and subsurface currents and lower covariances between subsurface currents and $\sigma_\theta$. The strong covariances between the surface and subsurface currents are also noticeable in P1 when the eddy surface observations are enough for its reconstruction in the water column. However, in that case, the eddy becomes weaker as depth increases (i.e., when moving away from the surface observations). This shows that apart from the observations and the parameters used for the ROOI, the reconstruction is also dependent on the covariances, highlighting the importance of a good historical

dataset. On the other hand, the reconstructed anticyclonic flow in P3 agrees with the Vgeos observed by the deep glider (Figure 4b), which is dependent on the water properties of the whole water column. In fact, the reconstructed across-track currents along the deep glider trajectory agree well with the across-track Vgeos observed by the deep glider (Figure 8), thus providing a first validation of the results. The difference in the intensity between both currents is likely due to the fact that the reconstructed currents have a daily variability.

The reconstructed eddy shows diameters of around 15–30 km, which are slightly smaller than those found in the literature at the surface in the study area [5]. On the other hand, the associated inshore transport values range between 0.04 and 0.15 Sv, being anisotropic and weaker when oriented offshore (negative transports). The values obtained here have a similar order of magnitude compared with the across-shelf transports modeled by [6] to the north of our study area in the Bay of Biscay, which were computed across the 500 m isobath for the first 50 m.

The results of the reconstructions agree with the water column properties measured by the deep glider and with the characteristics of eddies found in the literature. Nevertheless, they are not validated with any other external data sources. Thus, although the ROOI shows promising skills for reconstructing eddy-like structures and for estimating transports in the study area, further analysis and validation with independent observations are needed to ensure a robust reconstruction of such structures.

When using the ROOI, having robust historical data is a prerequisite to any attempt of 3D reconstruction of current velocity fields in order to ensure reliable relationships between the variables we want to use in the reconstruction and to be able to reproduce the target processes or features we want to reconstruct. The sensitivity to the main parameters used in the ROOI must be analyzed carefully when applying the method to other study areas. The choice of the parameters also depends on the structure or features that require reconstruction and in the relative importance that should be given to each input variable. In addition, as mentioned, validation of the results with additional observations is necessary to assess the skill.

## 5. Conclusions

This study continues with previous efforts to analyze coastal eddies in the SE-BoB [5], shedding some light on the 3D characteristics of these structures that have been analyzed only from surface observations until now. A coastal mode-water eddy has been characterized in 3D and, despite showing slightly smaller scales, the general characteristics were similar to those found in the literature at open water regions nearby. The mode-water eddy was characterized by the joint analysis of multiplatform observations, highlighting the potential of this kind of approach for a better characterization of the different oceanic features, especially in coastal areas, where satellite observations present several limitations.

In addition, a 3D data reconstruction method was used to assess its capability to reconstruct eddy-like structures and estimate the associated cross-shelf transports in the upper 100 m. The results show that the method is able to reconstruct mesoscale eddies in different scenarios and that their associated cross-shelf transports provide reasonable first results. Therefore, the ROOI is here presented as a compelling tool for reconstructing coastal 3D circulation and transports. However, further validation with external data sources of current velocities or transport estimations would be valuable to ensure the robustness of the results.

In the future, the analysis of the main physical drivers responsible for the generation and evolution (e.g., migration, decay) of these recurrent coastal eddies would be interesting for better understanding the origin and progression of such structures in the study area. This could be carried out by means of numerical simulations and additional observations, for which ad hoc sampling strategies would be desirable. A better depiction of these types of structures would constitute an advance in the understanding of their impact on the surrounding ecosystem. Additionally, the ROOI approach could have several applications. The first application would be to reconstruct current velocity fields to study

3D coastal transports from a Lagrangian approach. This can have interesting ecological applications, such as for instance, the estimation of the residence areas of passive particles, such as marine litter at different layers, or for studying the distribution of eggs and larvae of different pelagic fish species. In fact, previous studies suggested that within the SE-BoB, the coincidence in space and time of SWODDIES with anchovy spawning could favor its recruitment by advecting eggs and larvae to off-shelf areas of lower predation risk but still productive enough to support them [42]. Another future application could be to use the ROOI operationally in near real time and at different depths in order to continuously reconstruct 3D current velocity fields from the HFR and mooring observations to complement the surface information for supporting the correct follow-up and integrated management of the SE-BoB.

**Supplementary Materials:** The following are available online at https://www.mdpi.com/2072-42 92/13/4/674/s1, SM1 (Figure S1): Input variables for the reconstruction, SM2 (Figures S2–S14): Sensitivity tests of the reconstruction.

**Author Contributions:** Conceptualization, I.M.-N., A.R., G.J., J.C., L.M. and A.C.; writing—original draft, I.M.-N., A.R. and A.C.; writing—review and editing, I.M.-N., A.R., G.J., J.C., L.M. and A.C.; data curation, J.C. and L.M.; formal analysis and visualization, I.M.-N. All authors have read and agreed to the published version of the manuscript.

**Funding:** This study has been supported by the JERICO-NEXT and JERICO-S3 projects, funded by the European Union's Horizon 2020 Research and Innovation Program under grant agreement no. 654410 and 871153. This study has also been undertaken with the financial support of the Department of Environment, Regional Planning, Agriculture and Fisheries of the Basque Government (Marco Program). We also acknowledge the Helmholtz Association for funding from the PACES II and PoF-IV programmes. Ivan Manso-Narvarte was supported by a PhD fellowship from the Department of Environment, Regional Planning, Agriculture and Fisheries of the Basque Government.

**Data Availability Statement:** The datasets used in this study are publicly available at the following links:
- Deep glider: http://www.ifremer.fr/co/ego/ego/v2/comet/comet_20180517/ (accessed on 9 December 2020)
- Shallow glider: http://www.ifremer.fr/co/ego/ego/v2/sebastian/sebastian_20180517/ (accessed on 9 December 2020)
- HFR: https://www.euskoos.eus/en/data/basque-ocean-meteorological-network/high-frequency-coastal-radars/ (accessed on 9 December 2020)
- Mooring: https://www.euskoos.eus/en/data/basque-ocean-meteorological-network/donostia-deep-water-buoy/ (accessed on 9 December 2020)
- Chl-a images: https://resources.marine.copernicus.eu/?option=com_csw&view=details&product_id=OCEANCOLOUR_ATL_CHL_L3_NRT_OBSERVATIONS_009_036 (accessed on 9 December 2020)
- Wind: http://mandeo.meteogalicia.es/thredds/catalogos/DATOS/ARCHIVE/WRF/WRF_hist.html (accessed on 9 December 2020)
- IBI: https://resources.marine.copernicus.eu/?option=com_csw&view=details&product_id=IBI_MULTIYEAR_PHY_005_002 (accessed on 9 December 2020)

**Acknowledgments:** We thank the Emergencies and Meteorology Directorate (Security department) of the Basque Government for public data provision from the Basque Operational Oceanography System EuskOOS. This study has been conducted using EU Copernicus Marine Service information. Wind data were obtained from the meteorological agency of Galicia (MeteoGalicia). Glider data were provided by the Everyone's Gliding Observatories (EGO) project. This is contribution number 1019 of the Marine Research Division of AZTI-BRTA.

**Conflicts of Interest:** The authors declare no conflict of interest.

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
