# Peer review of "Three-Dimensional Characterization of a Coastal Mode-Water Eddy from Multiplatform Observations and a Data Reconstruction Method"

_remotesensing, doi:10.3390/rs13040674_

Round 1

Reviewer 1 Report

The authors presented a multiplatform approach that combines observations from high-frequency radar, satellite and gliders, aiming to characterize three-dimensional structure of a coastal mode-water eddy. Moreover, the eddy current velocity was reconstructed through a three-dimensional data reconstruction method. The methods are well presented. The results are clear.

Reviewer 2 Report

The manuscript entitled “Three-dimensional characterization of a coastal mode-water eddy from multiplatform observations and a data reconstruction method,” highlights a multiplatform data approach for characterizing coastal mesoscale processes.  The paper is well-written and timely as our field moves more too autonomous data collections and instrumentation.  After reading through the work several times, I am hard pressed to find any major short-comings with the work.  While I am not overly familiar with the coastal mesoscale literature, I concur that the work is novel in its application of multiplatform data (glider, mooring, HF, satellite) to the coastal surface/subsurface characterization problem.  My only minor comment would be a possible paragraph discussing optimal in situ sampling strategies given their findings?  For example, if you were developing an in situ observational system to support remote sensing observations to characterize coastal eddies, what would it look like with respect to spatial/temporal path planning and sampling resolution?  Would additional persistent autonomous platforms help resolve the physical features of an eddy (i.e. multiple Slocum style gliders; surface Wave Gliders from Liquid Robotics)?  Given my review of the work, I recommend the manuscript be published in Remote Sensing with minor revisions.

Reviewer 3 Report

This article focuses on three-dimensional characterization of a coastal mode-water eddy. Their description and characterization are challenging. Because observations are too scarce for studying physical properties, and environmental impacts at the required spatio-temporal resolution. This article presents multiplatform observations and a data reconstruction method. The mode-water eddy was characterized by the joint analysis of multiplatform observations. The paper highlights the potential of this kind of approach for a characterization of the different oceanic features, especially in coastal areas, where satellite observations present several limitations. The method of data reconstruction is able to reconstruct mesoscale eddies in different scenarios and their associated cross-shelf transports provide reasonable first results.

Some particular points to improve: (see the commented pdf file for more details):

  1. Line33-35, Keywords are too long. Try to reduce.
  2. This paper uses the measurement data obtained by various sensors from multiplatformobservations (high-frequency radar, satellite and 2 gliders), and the principles and accuracy of the data obtained by different sensors are different. Please explain in detail how the data measured by different sensors from multiplatform observations (high-frequency radar, satellite and 2 gliders) are fused. What is the principle, and whether multi-source data fusion improves the accuracy of reconstructing eddy?
  3. How the data reconstruction method proposed in this paper can reflect the authenticity of the reconstruction eddy, please use some indicators or formulas to demonstrate.
  4. Table 1: Author gave a series of values about details of the different observing platforms and data sets used are summarized. It would be better if the author could analyze the methods in the table such as advantages and disadvantages.
  5. Line85-97, Line107-110, Line306-314, Line378-388: The author’s description of the figures and tables in the paper is too redundant. Please highlight the key content and put less unnecessary details.
  6. It is particularly persuasive if authors can compare their own approach to existing ones. There is a lack of effective comparative analysis with previous research methods in this paper.

Round 2

Reviewer 3 Report

The manuscript has been revised carefully according to the comments of the reviewer. It is suggested to correct the following two questions in the manuscript and then be accepted.

  1. Good sampling strategy, help to characterize eddies in the study area well. For example, the author proposed that an improved strategy can be based on the deployment in the area covered by the HFR of two deep water gliders that sample the eddy core and periphery and also cross the slope and shelf-break areas in perpendicular directions. However, is the 3D data reconstruction method still applicable with the change of the sampling strategy in this paper? please use formula to prove.
  2. A more careful account of the introduction is recommended. Please clearly state the advantages of your research compared with predecessors, the innovation of the method and the engineering application value.
  3. The tables and figures in the paper are not very standardized and readable, such as Table 1. Please modify them one by one.
